# Seasonal Variations in the Structure and Function of the Gut Flora in Adult Male Rhesus Macaques Reared in Outdoor Colonies

**DOI:** 10.3390/microorganisms13010117

**Published:** 2025-01-08

**Authors:** Longbao Lv, Feiyan Zhang, Haimei Zhou, Wenxian Xiao, Yingzhou Hu, Wenchao Wang, Zhu Zhu, Fangming Zhu, Dongdong Qin, Xintian Hu

**Affiliations:** 1University of Chinese Academy of Sciences, Beijing 101408, China; lvlongbao@mail.kiz.ac.cn; 2National Resource Center for Non-Human Primates, and National Research Facility for Phenotypic & Genetic Analysis of Model Animals, Kunming Institute of Zoology (Primate Facility), Chinese Academy of Sciences, Kunming 650107, China; zhangfeiyan@mail.kiz.ac.cn (F.Z.); xiaowenxian@mail.kiz.ac.cn (W.X.); 3Key Laboratory of Traditional Chinese Medicine for Prevention and Treatment of Neuropsychiatric Diseases, Yunnan University of Chinese Medicine, Kunming 650500, China; 17773572364@163.com; 4Key Laboratory of Animal Models and Human Disease Mechanisms of the Chinese Academy of Sciences & Yunnan Province, and KIZ-CUHK Joint Laboratory of Bioresources and Molecular Research in Common Diseases, Kunming 650201, China; huyingzhou@mail.kiz.ac.cn (Y.H.); wangwenchao@mail.kiz.ac.cn (W.W.); zhuzhu1@mail.kiz.ac.cn (Z.Z.); zhufangming@mail.kiz.ac.cn (F.Z.)

**Keywords:** rhesus monkey, 16S rRNA sequencing, gut microbiota, seasonal variation

## Abstract

The seasonal variations that occur in the gut microbiota of healthy adult rhesus monkeys kept in outdoor groups under conventional rearing patterns and how these variations are affected by environmental variables are relatively poorly understood. In this study, we collected 120 fecal samples from 30 adult male rhesus monkeys kept in outdoor groups across four seasons and recorded the temperature and humidity of the housing facilities, as well as the proportions of fruit and vegetables in their diet. A 16S rRNA sequencing analysis showed that the alpha diversity of the gut microbiota of the rhesus monkeys was higher in winter and spring than in summer and autumn. A principal coordinate analysis (PCoA) further demonstrated notable seasonal variations in the composition and functionality of the gut microbiota in the rhesus monkeys. The phyla Firmicutes and Bacteroidetes and the genus *Prevotella 9* were the significantly dominant groups in all 120 fecal samples from the rhesus monkeys. A linear discriminant analysis (LDA) effect size (LEfSe) analysis (LDA > 4) indicated that at the phylum level, Firmicutes was significantly enriched in winter, Bacteroidetes was significantly enriched in summer, and Proteobacteria and Campylobacter were significantly enriched in spring. At the genus level, *Helicobacter* and *Ralstonia* were significantly enriched in spring; *Prevotella 9*, *Streptococcus*, and *Prevotella* were significantly enriched in summer; and UCG_005 was significantly enriched in autumn. The beneficial genera *Lactobacillus*, *Limosilactobacillus*, and *Ligilactobacillus* and the beneficial species *Lactobacillus johnsonii*, *Limosilactobacillus reuteri*, *Ligilactobacillus murinus*, and *Lactobacillus amylovorus* all showed the same seasonal trend; namely, their average relative abundance was markedly greater during the winter months compared to other seasons. Compared with other seasons, carbohydrate metabolic function was significantly upregulated in winter (*p* < 0.01), amino acid metabolic function was relatively increased in spring, and energy metabolic function and the metabolic function of cofactors and vitamins were significantly downregulated in winter and relatively upregulated in summer. A variance partitioning analysis (VPA) and redundancy analysis (RDA) showed that the proportions of fruits and vegetables in the diet, but not climatic factors (temperature and humidity), significantly influenced the seasonal changes in the gut microbiota. These variations were related to changes in the proportions of fruits and vegetables. This research presents novel findings regarding the influence of external environmental factors on the gastrointestinal environment of rhesus monkeys.

## 1. Introduction

The digestive tracts of animals harbor vast bacterial ecosystems that are in a delicate balance with their host environments. The host relies on the microbial community to degrade dietary fibers and other food components, thus playing a vital role in the host’s metabolism, immunity, and defenses against pathogens, among other physiological activities [1,2,3]. Conversely, the composition of the gut microbiota is influenced by factors such as host genetics, diet, seasons, age, social environment, antibiotic use, digestive tract structure, and growth stage [4,5,6]. Seasonal fluctuations in the composition of gut microbiota in animals are primarily influenced by seasonal changes in dietary intake [7,8]. Moreover, several longitudinal studies have suggested that the composition of the gut microbiota is not constant over time and is affected by environmental variations. In mammals like black howler monkeys, white-faced capuchins, plateau pikas, and musk oxen, the gut microbiome demonstrates seasonal fluctuations [7,9,10,11,12], as do those of many fishes [13], amphibians [14], and insects [15]. The composition of the gut microbiota of non-human primates is significantly affected by lifestyle disruptions, such as long-term captive breeding, which represents a dietary alteration compared with wild populations [16]. While seasonal changes in the composition of the gut microbiome of animals are mainly driven by seasonal changes in diet, they can also be indirectly influenced by seasonal climatic fluctuations, such as changes in humidity and air temperature [17,18,19].

Rhesus macaques are closely related to humans [20]. Indeed, 93% of the genes in rhesus macaques exhibit homology with the human genome, and both species possess analogous tissue architectures, along with comparable physiological and metabolic traits. Rhesus macaques have been extensively utilized in investigations concerning cerebral function, cognitive processes, neuropsychiatric disorders, stem cell research, mechanisms of infectious diseases, and the advancement of vaccines [21]. With the advancement of biological and medical research, studies increasingly rely on the use of non-human primates, such as the rhesus macaque. The artificial breeding of rhesus macaques not only provides experimental animals with clear genetic backgrounds and stable physiological and biochemical indexes for medical biological research but also plays an important role in the protection of the ecological environment. It is crucial to formulate suitable dietary plans and create appropriate environmental conditions in the artificial breeding of rhesus macaques. In artificial feeding, the diet usually consists of compound feed, along with fruits and vegetables. Given that the compound feed is generally fixed, the choice and proportion of the fruit and vegetable feed assume special relevance. Monkeys are known to prefer food that is slightly sweet and soft, and their intestines are well-suited to digesting foods rich in carbohydrates [22]. The main fruit in their diet is apples, while the vegetable component primarily comprises root vegetables, such as sweet potatoes, pumpkins, cabbage, kohlrabi, white radishes, and onions, which, together, meet the nutritional needs and dietary habits of the macaques. However, how to scientifically and reasonably balance fruit and vegetable proportions is unclear and requires further in-depth exploration. In artificial feeding and breeding, in addition to optimizing the dietary structure, it is equally important to maintain the appropriate environmental temperature and humidity. Both low and high temperatures can affect the reproductive function of the macaques [23]. Under high humidity, evaporation from the monkey’s body surface is inhibited, which frequently results in metabolic disorders, diminished resistance, and increased disease incidence. The incidence of intestinal diseases in macaques is relatively high during seasonal changes, with bacterial dysentery ranking first [24]. Some studies have suggested that the insufficient absorption of folic acid and vitamin C from food can result in reduced resistance to dysentery, highlighting the need to provide rhesus monkeys with green fodder and fruits to enhance their resistance to this intestinal disease. The gut microbiome serves as a biological indicator of the health and welfare of rhesus monkeys. Nevertheless, how the seasonal changes in the intestinal flora of outdoor, group-housed rhesus monkeys maintained under conventional feeding patterns are affected by the proportion of fruits and vegetables provided or by temperature and humidity remains unclear. Therefore, discerning the changes in the diversity and composition of the gut microbiome of outdoor, group-housed macaques through the different seasons is crucial for assessing the suitability of the current feeding environment and diet, as well as for understanding whether these changes can affect the health of the animals.

In this study, we monitored 30 outdoor, group-housed adult male rhesus monkeys aged 10 to 15 years, and their weights were measured at a standardized time during each season. Feces were sampled from 30 individuals, and the ratios of fruits and vegetables and the climatic conditions at the sampling site were surveyed over four seasons. Fecal microbiota profiling was conducted using 16S rRNA sequencing. Alpha diversity, dominant bacterial taxa, and microbial function were compared among the different seasons. The influence of environmental variables on gut microbiota composition was analyzed. Ultimately, seasonal fluctuations in gut microbial functionalities and the relationships between gut microbial biomarkers and environmental factors were examined. The aim of this work was to identify differences in the composition and functional profiles of the gut microbiota in rhesus monkeys across the four seasons and determine the contribution of the proportions of fruits and vegetables provided in the diet to these differences. This study not only provides important information for understanding the seasonal changes and environmental adaptations in the gut microbiota in adult rhesus macaques raised in outdoor groups under artificial feeding conditions but also forms a basis for optimizing the selection and proportion control of fruits and vegetables in artificial diets. Our findings further address gaps in current domestic regulations and standards for laboratory primate feeds, which are limited to requirements for compound feeds, fruits and vegetables, and green fodder. As such, they lack objective evaluation criteria for the inclusion of fruits and vegetables in the diets of rhesus macaques, which may negatively affect the health and welfare of these animals.

## 2. Materials and Methods

### 2.1. Animal Selection

Animals were chosen based on the timetable for the forthcoming routine semiannual evaluations. Medical histories were examined, and subjects were excluded based on any of the following criteria: positive tests for simian retrovirus D (SRV), simian immunodeficiency virus (SIV), simian T-lymphotropic virus 1 (STLV-1), cercopithecine herpesvirus type 1 (BV), and *Mycobacterium tuberculosis* (TB), as well as *Salmonella* and *Shigella* infection, history of prior diarrhea episodes, recent administration of antibiotics within the last 90 days prior to sample collection, and additional clinical comorbidities. A total of thirty healthy adult male rhesus macaques were assessed as meeting the criteria for inclusion. Throughout the experiment, daily health checkups were performed by veterinarians. None of the 30 macaques exhibited diarrhea or other clinical symptoms, and none received any medication or probiotics. Their weights were recorded once each season at a consistent time.

### 2.2. Animal Housing

The macaques were kept in the Laboratory Animal Center of Kunming Institute of Zoology, Chinese Academy of Sciences (Kunming Yunnan, China). The housing conditions and animal care were approved by the China National Accreditation Service for Conformity Assessment (CNAS). The animals were given standard hygiene practices, a sufficient and consistent nutritional regimen, and a stable social environment. All macaques were housed in an outdoor monkey house and were not subjected to unnecessary manual intervention. Daily health checkups were performed by veterinarians. The experiments were conducted at the Laboratory Animal Center of Kunming Institute of Zoological Research, Chinese Academy of Sciences [SYXK(Dian)K2022-0009], and this study was approved by the Ethics Committee for Animal Experiments of Kunming Institute of Zoological Research, Chinese Academy of Sciences (Ethics Approval No.PE-2021-07-002). The experiments were conducted in accordance with the principles of the 3Rs.

### 2.3. Fecal Sample Collection

From December 2022 to September 2023, rectal swabs were collected from each of the 30 adult macaques, once in each of the four seasons. All the swabs were marked at 2.2 cm from the tip and then inserted into the rectum up to the marked line. The collected samples were placed in cryovials, rapidly frozen in liquid nitrogen, and stored at −80 °C for testing. The collections were conducted in March (spring), June (summer), September (autumn), and December (winter).

### 2.4. Environmental Factors

The Laboratory Animal Center of the Kunming Institute of Zoology, Chinese Academy of Sciences, is located in Huahongdong, in the western suburbs of Kunming Yunnan, China, at an elevation of approximately 2186 m. The temperature and humidity in the breeding facilities are monitored using environmental sensors. The diet of the monkeys consisted of compound feed and fruit and vegetable feed at a ratio of 45% to 55%, respectively. All macaques were given the same compound feed with the following nutritional composition: crude protein 20.97%, crude fat 5.50%, crude fiber 3.30%, crude ash 6.20%, calcium 1.00%, and phosphorus 0.63%. The fruit and vegetable feed included seven types of food that were uniformly diced and mixed in different proportions according to the season—winter: 82% apple, 6% sweet potato, 0% pumpkin, 6% cabbage, 3% kohlrabi, and 3% white radish + onion; spring: 78% apple, 6% sweet potato, 0% pumpkin, 6% cabbage, 6% kohlrabi, and 4% white radish + onion; summer: 70% apple, 7% sweet potato, 1% pumpkin, 6% cabbage, 10% kohlrabi, and 6% white radish + onion; and autumn: 62% apple, 11% sweet potato, 3% pumpkin, 7% cabbage, 7% kohlrabi, and 10% white radish + onion.

### 2.5. DNA Extraction and 16S rRNA Gene Sequencing

Genomic DNA from microbial communities was extracted from all fecal samples utilizing the E.Z.N.A. soil DNA Kit (Omega Bio-Tek, Norcross, GA, USA) in accordance with the manufacturer’s protocol. The concentration and purity of the DNA were evaluated using 1% agarose gel, and the DNA was subsequently diluted to 1 ng/μL with sterile water. The V3-V4 hypervariable region of the bacterial 16S rRNA gene was amplified using the primer pair 341F (5′-CCTAYGGGRBGCASCAG-3′) and 806R (5′-GGACTACHVGGGTWTCTAAT-3′) on a Bio-Rad T100 gradient PCR thermocycler (BioRad, Hercules, CA, USA). PCR reactions were conducted in a 30 μL volume comprising 15 μL of Phusion High-Fidelity PCR Master Mix (New England Biolabs, Ipswich, MA, USA), 0.2 μM each of forward and reverse primer, and 10 ng of template DNA. The thermal cycling protocol included an initial denaturation at 98 °C for 1 min, followed by 30 cycles of denaturation at 98 °C for 10 s, annealing at 50 °C for 30 s, and elongation at 72 °C for 30 s, concluding with a final extension at 72 °C for 5 min. Equal volumes of 1× loading buffer (containing SYB green) and PCR products were combined and subjected to 2% agarose gel electrophoresis for band visualization. The PCR products were pooled in equimolar ratios and purified using a GeneJET Gel Extraction Kit (Thermo Fisher Scientific, Waltham, MA, USA). Sequencing libraries were constructed using an NEB Next Ultra DNA Library Prep Kit for Illumina (New England Biolabs, Ipswich, MA, USA) following the manufacturer’s guidelines, with index codes incorporated. Library quantification and a quality assessment were performed using a Qubit 2.0 Fluorometer (Thermo Fisher Scientific, Waltham, MA, USA) and an Agilent Bioanalyzer 2100. Ultimately, the library was sequenced on the Illumina HiSeq platform, producing 250 bp paired-end reads (Novogene Co., Ltd., Beijing, China).

### 2.6. Data Analysis

#### 2.6.1. Data Quality Control

Paired-end reads were allocated to samples according to their distinct barcodes, followed by the truncation of barcode and primer sequences. This entire procedure was executed in Python (v.3.6.13), with the removal of adapters and primer sequences accomplished using cutadapt (v.3.3). The reads were subsequently merged utilizing FLASH (v.1.2.11, http://ccb.jhu.edu/software/FLASH/) (accessed on 28 February 2024) [25], resulting in the generation of raw reads. The quality filtering of these raw reads was conducted using fastp (v.0.23.1) software to yield high-quality, clean tags [26]. These tags were then cross-referenced against established databases (SILVA database [16S], https://www.arb-silva.de/; (accessed on 3 March 2024) Unite database [ITS], https://unite.ut.ee/) (accessed on 3 March 2024) to identify chimeric sequences. Effective tags were obtained by eliminating chimeric sequences with the vsearch package (v.2.16.0, https://github.com/torognes/vsearch) (accessed on 4 March 2024).

#### 2.6.2. Bioinformatics Analysis

All data analyses were carried out on the NovoMagic cloud-based platform (https://magic.novogene.com/customer/main#/homeNew) (accessed on 7 August 2024). In summary, all effective tags underwent denoising via DADA2, producing Amplicon Sequence Variants (ASVs). Species annotation was subsequently performed based on the clustered ASV sequences utilizing the SILVA database (release 138.1). Based on the ASV clustering outcomes, abundance (expressed as mean ± standard deviation) and alpha and beta diversity indices were computed using Qiime 1.9.1 software.

A principal coordinate analysis (PCoA) was employed to evaluate beta diversity among the various groups using the “vegan” package in R software 4.1.0. The Bray–Curtis distance metric was utilized to assess similarities between samples at the operational taxonomic unit (OTU) level. An analysis of similarities (ANOSIM), a non-parametric statistical test, was applied to examine intergroup differences using R software 4.1.0 (“anosim” function in the “vegan” package). The metabolic functions of the bacterial communities were predicted using Phylogenetic Investigation of Communities by Reconstruction of Unobserved States (PICRUSt) software (1.1.4), based on the Kyoto Encyclopedia of Genes and Genomes (KEGG) databases, utilizing OTU species annotation and abundance data. Furthermore, the Wilcoxon rank sum test was employed to analyze seasonal variations in metabolism-related functions and the predominant bacteria across all groups. The linear discriminant analysis effect size (LEfSe) was utilized to identify taxa with significant differences in abundance across the four seasons at various taxonomic levels. Statistical significance was determined using an LDA score exceeding 4 and a *p*-value below 0.05. A distance-based linear model, redundancy analysis (RDA), and variance partitioning analysis (VPA) were utilized to assess the impact of external factors on the gut microbiota, while Spearman’s correlation was applied to investigate the interrelationships between species and environmental factors. The chi-squared test and Fisher’s exact test were employed to evaluate differences in the unique microbiota.

## 3. Results

### 3.1. Animal Demographics

Thirty healthy adult male rhesus macaques are negative for seven pathogens (TB, SRV, STLV, SIV, BV, *Salmonella* and *Shigella*). These animals have never had diarrhea and have not used any medication or probiotics. Their ages range from 10 to 15 years (11.50 ± 1.78). The mean body weight was 11.35 ± 2.27 kg in winter, 10.57 ± 2.31 kg in spring, 10.47 ± 2.42 kg in summer, and 10.87 ± 2.62 kg in autumn. No significant differences were found in body weights among the four seasons (*p* > 0.05) (Figure 1).

### 3.2. Assessment of Sequencing Data

A total of 9,647,634 high-quality clean reads (80,397 per sample) were obtained in all the samples. The rarefaction curves for Sobs and Shannon indexes at the OTU level gradually leveled off with increasing sequencing depth (Figure 2). These results indicated that the sequencing depth was sufficient, as each fecal sample had enough OTUs to reflect the maximum level of bacterial diversity.

Following DADA2-based denoising, the sequences were clustered at 100% similarity, resulting in a total of 5320 ASVs. These ASVs were annotated against the SILVA database (release 138), leading to the identification of 39 phyla, 78 classes, 182 orders, 309 families, and 666 genera.

### 3.3. Gut Microbiota Composition in Adult Rhesus Macaques Across Seasons

At the phylum level, Bacteroidetes and Firmicutes were dominant in different seasons, with average relative abundances of 42.85 ± 16.13% and 41.80 ± 18.09%, respectively, followed by Campylobacterota (3.66%) and Spirochaetota (1.80%), which were also dominant (relative abundance > 1%) (Figure 3A). The other six phyla were Proteobacteria, Actinobacteria, Desulfobacterota, Fusobacteriota, Acidobacteriota, and Verrucomicrobia. The relative abundance of the top 10 phyla accounted for the majority (99%) of the detectable reads in all samples. At the genus level, *Prevotella 9* (29.09%) and *Ralstonia* (4.74%) showed the greatest relative abundance. The other genera displaying average relative abundances greater than 1% (Figure 3B were *Faecalibacterium* (3.90%), *Streptococcus* (3.82%), *Helicobacter* (3.43%), *Prevotella* (3.43%), *Ligilactobacillus* (3.08%), *Lactobacillus* (2.77%), UCG-002 (2.51%), *Alloprevotella* (2.04%), *Limosilactobacillus* (2.18%), UCG-005 (1.98%), and *Treponema* (1.22%) (Figure 3B). In summary, the dominant genera in rhesus monkeys across the four seasons belonged to the phyla Firmicutes and Bacteroidetes, with *Prevotella 9* being a key microorganism closely related to a diet rich in dietary fiber.

### 3.4. Analysis of the Seasonal Differences in the Gut Microbiota

#### 3.4.1. Seasonal Variation in Alpha Diversity

Shannon and Chao1 indices were employed to evaluate alpha diversity, while the Wilcoxon rank sum test was used to assess the significance of seasonal differences. We noted that the Shannon index of rhesus monkeys was significantly lower in summer than in autumn and winter (*p* < 0.001) (Figure 4A). Meanwhile, the Chao1 index was significantly higher in spring than in summer and autumn (*p* < 0.05) (Figure 4B). Overall, the alpha diversity of the gut microbiota in rhesus monkeys was higher in winter and spring than in autumn and summer.

#### 3.4.2. Seasonal Variations in Beta Diversity

The Bray–Curtis distance algorithm was used to calculate the differences among rhesus monkey fecal samples in different seasons. ANOSIM was applied to test whether the differences among samples from different seasons were significantly greater than those within the same season. A PCoA analysis showed that samples from different seasons formed distinct clusters, while samples from the same season clustered closely together, with a clear separation between winter and summer samples. This indicated that there were significant seasonal variations in the composition of the gut microbiota of rhesus monkeys (R = 0.4728, *p* = 0.001) (Figure 4C). The ANOSIM results indicated that the differences among samples from different seasons were significantly greater than the differences among samples from the same season, with significant differences between summer and winter (R = 0.806, *p* = 0.001) and between spring and winter (R = 0.5899, *p* = 0.001) being noted (Table 1).

#### 3.4.3. Seasonal Variations in Microbial Communities

An LEfSe analysis (LDA threshold of 4) was used to identify taxa showing significant seasonal variations in abundance, with four groups being identified at the phylum level. Compared with other seasons, Firmicutes were significantly enriched in winter (LDA > 4, *p* < 0.05), Bacteroidetes were significantly enriched in summer, and Proteobacteria and Campylobacter were significantly enriched in spring. At the genus level, nine genera showed marked differences in relative abundance. Among these, *Lactobacillus*, *Limosilactobacillus*, and *Ligilactobacillus* were enriched in winter; *Helicobacter* and *Ralstonia* were enriched in spring; *Prevotella 9*, *Streptococcus*, and *Prevotella* were enriched in summer; and UCG_005 was enriched in autumn. At the species level, eight species showing significant differences were identified. These species included *L. johnsonii*, *L. reuteri*, *L. murinus, L. amylovorus*, and *L. salivarius*, which were enriched in winter; *Helicobacter fennelliae* and *Ralstonia pickettii*, which were enriched in spring; and *Streptococcus lutetiensis*, which was enriched in summer (Figure 5). Based on the Wilcoxon rank sum test, the ratio of Firmicutes to Bacteroidetes (F/B ratio) was significantly higher in winter than in other seasons.

### 3.5. Analysis of Beneficial and Harmful Microorganisms

An LEfSe analysis was used to compare the abundances of the differential taxa across different seasons. The beneficial bacterial genera (*Lactobacillus*, *Limosilactobacillus*, and *Ligilactobacillus*) showed consistent seasonal changes (Figure 6A–C), with a significantly higher average relative abundance in winter than in other seasons. The species *L. johnsonii*, *L. reuteri*, *L. murinus*, and *L. amylovorus* showed the same seasonal variation trend (Figure 6D–G). The main genera of conditionally pathogenic bacteria present in the intestinal tract of rhesus macaques are *Streptococcus* and *Helicobacter*. Here, we observed that the average relative abundance of *Helicobacter* in spring was higher than in other seasons (Figure 6H), while that of *Streptococcus* was higher in summer than in other seasons (Figure 6I).

In the feeding and management of rhesus macaques in this study, *Salmonella*, *Shigella*, and *Clostridium difficile*, common diarrhea-causing pathogenic bacteria in rhesus macaques, were not detected within the macaque population. The main pathogenic genera identified in the population were *Pseudomonas*, *Campylobacter*, *Vibrio*, and *Escherichia–Shigella*. *Campylobacter* was significantly more frequently detected in winter (100%, 30/30) than in autumn (100%, 30/30). Pseudomonas was significantly more prevalent in winter (100%, 30/30) and spring (80%, 24/30) than in summer (10%, 3/30) and autumn (6.67%, 2/30) (Table 2).

### 3.6. The Effects of Environmental Factors on the Seasonal Variation in the Gut Microbiota of Rhesus Macaques

The temperature was significantly higher in summer and autumn than in spring and winter, while the humidity was significantly higher in autumn and winter than in spring and summer (Figure 7A). The highest percentages of apples, cabbage, sweet potatoes, and white radishes with onions were provided in winter, summer, and autumn, respectively, and pumpkins were not given in winter and spring (Figure 7B).

A variance partitioning analysis (VPA) was conducted to assess the relative impacts of various environmental factors, including climatic and dietary influences, on alterations in the bacterial community composition utilizing the “varpart” method. Climatic factors (temperature and humidity) alone explained 0% of the observed variation, while dietary factors (fruits and vegetables) alone explained 2.58% of the observed variation. A combination of both factors explained 20.24% of the observed variation (Figure 7C). This indicated that while these climatic factors alone could not explain the changes in the ecological community, they made a significant contribution to the observed changes when combined with dietary (fruit and vegetable-related) factors. The results of the RDA analysis also showed that temperature (r^2^ = 0.84, *p* = 0.58) and humidity (r^2^ = 0.99, *p* = 0.13) were not significantly correlated with the microbial community. Spearman’s correlation analysis was conducted to determine the relationship between the types of fruits and vegetables and the relative abundances of the gut microbiota (Figure 7D). The 10 most dominant genera were selected for this assessment. Apples displayed a positive correlation with the relative abundance of the genera *Lactobacillus*, *Limosilactobacillus*, and *Ligilactobacillus* and a negative correlation with the relative abundance of the genus *Prevotella 9*. The abundance of *Prevotella 9* showed a positive correlation with kohlrabi, pumpkin, and white radish with onion.

### 3.7. Seasonal Differences in Gut Microbial Functions

A principal component analysis (PCA) was performed based on the statistical analysis of abundance from database-based functional annotations. The results showed that samples from different seasons formed separate clusters, while samples from the same season clustered together (Figure 8A). A clustering heatmap based on the relative abundance in level 2 KEGG pathways showed that monkeys that were not fed pumpkin but were fed a high proportion of apples in winter and spring were clustered into one group, while those that received fewer apples in summer and autumn were clustered into another group (Figure 8B). A functional prediction analysis conducted on the KEGG database showed that the gut microbiota in outdoor-housed rhesus monkeys was mainly associated with membrane transport, translation, replication and repair, carbohydrate metabolism, amino acid metabolism, energy metabolism, and cofactor and vitamin metabolism (Figure 8C). The KEGG analysis showed that the magnitude of the four major metabolism-related functions of the gut microbiota of rhesus monkeys displayed seasonal differences. Overall, carbohydrate metabolism was significantly higher in winter than in the other seasons (*p* < 0.01), while amino acid metabolism was relatively higher in spring than in the other seasons. Energy metabolism and cofactors and vitamin metabolism were significantly lower in winter and relatively higher in summer than in spring or autumn (Figure 8D). The gut bacteria of the rhesus monkeys shared 4574 KEGG Orthology (KO) terms over the seasons. In total, 87, 205, 13, and 40 KO terms were unique to December, March, June, and September, respectively (Figure 8E).

## 4. Discussion

In this study, Firmicutes and Bacteroidetes were the significantly dominant phyla (relative abundance > 90%) among the 120 fecal samples obtained during the four seasons from rhesus macaques raised in large outdoor cages, which was consistent with previous reports [27,28]. Firmicutes and Bacteroidetes have been identified as the predominant phyla in non-human primates, comprising proportions ranging from 70.50% to 98.30%, respectively [4]. Firmicutes facilitate the breakdown of dietary fiber and the conversion of cellulose into volatile fatty acids, thereby improving digestive efficiency and supporting growth and development [29]. In our investigation, we found that within the top 30 bacterial genera present in the gut microbiota of rhesus macaques, 10 were classified as Firmicutes, which is linked to carbohydrate metabolism, as well as cellulose digestion and absorption. Furthermore, seven genera were categorized within the phylum Bacteroidetes, whose members are primarily involved in the digestion and absorption of proteins and carbohydrates and contribute to the maturation of the gastrointestinal immune system. Studies have indicated that a diet rich in fiber can enhance the abundance of *Prevotella* in the gut [30]. Most of the seven above-mentioned genera were *Prevotella*, which may be related to the fact that the dietary structure of the rhesus macaque diets in this study was plant-based (fruits and vegetables). This dietary pattern resembles the Mediterranean diet (MD) in humans and is also a plant-based diet. The MD is associated with the genus *Prevotella* and with higher levels of *Faecalibacterium prausnitzii*, increasing short-chain fatty acid (SCFA)-producing bacteria [31]. The microbiota plays a crucial role in deriving energy and nutrients from plant-based diets, potentially aiding primates in adapting to new dietary niches as a response to swift environmental changes [32].

At the genus level, the dominant genera in macaques are primarily those that produce SCFAs [33], such as *Prevotella 9*, *Streptococcus*, *Faecalibacterium*, *Prevotella*, *Ligilactobacillus*, *Lactobacillus*, UCG-002, and UCG-005. These SCFAs may play a role in decreasing the incidence of malignancies, such as colorectal cancer, and enhancing cardiometabolic health. Also, these SCFAs can be involved in the regulation of immune function, glucose and lipid metabolism, as well as blood pressure, all of which are linked to the Mediterranean diet (MD) [34,35]. Research has shown that *Faecalibacterium* is a butyrate-producing bacterium that exerts a positive impact on human energy metabolism and possesses good anti-inflammatory characteristics [36]. UCG002 and UCG005 digest high-fiber foods and function as symbiotic bacteria in the gut [37]. The beneficial genera *Ligilactobacillus*, *Lactobacillus*, and *Limosilactobacillus* can inhibit the excessive growth of conditional pathogens in the gut by competing with them for adhesion sites, as well as producing bacteriocins, organic acids, hydrogen peroxide, and other antimicrobial substances [38]. Their metabolic products are of great interest for their ability to regulate immune function, in addition to their anti-tumor activity, antibacterial properties, and preservative functions. This study also includes pathogenic bacteria of the genus *Helicobacter* and *Ralstonia*. *Helicobacter* is associated with diarrhea in rhesus monkeys and can cause disease under specific conditions, such as when the immune system is weakened or Helicobacter infection levels are high [39]. Additionally, *Ralstonia* was detected in the rectal swabs of rhesus monkeys in our study, which aligns with the work of [40], which showed that this genus is present in the rectum of these macaques. Studies have indicated that this genus, which is widely found in nature, is an emerging pathogen, particularly in water sources, and can survive under poor nutritional conditions [11]. During the daily care and management of rhesus macaques, exposure to *Ralstonia* through the handling or ingestion of contaminated toys or food might allow the microbes to enter and colonize the macaque intestine. Accordingly, it is essential to strengthen the cleaning, disinfection, and hygiene management of fruits and vegetables in rhesus macaque rearing environments to reduce the health risks of potential pathogens such as *Ralstonia*.

PCoA analyses revealed that in rhesus monkeys, the gut microbial structure and composition vary significantly according to season, which is consistent with other reports [41,42]. Seasonal differences in the gut microbiome are closely related to food resources, dietary structure, nutrient utilization, and feeding patterns [43,44]. Here, we found that the alpha diversity of the gut microbiota exhibited marked seasonal variations in rhesus monkeys, as determined by the Wilcoxon rank sum test. Overall, the alpha diversity of the gut microbiota was higher in winter and spring than in autumn and summer. It has been shown that an increase in alpha diversity leads to a more complex and stable intestinal microbiota composition, thereby enhancing resistance to external interference and adaptability, which is beneficial to the host’s health [45]. Changes in alpha diversity are associated with a variety of diseases [46]. Therefore, the observed increase in alpha diversity in the gut microbiota of the rhesus monkey in winter and spring may improve the resistance to, and reduce the influence of, adverse environmental factors, as well as promote the intake of fiber-rich food and nutrient absorption and utilization in the cold season. Increased bacterial diversity in the gut microbiota in winter and spring may be due to lower temperatures, higher humidity levels, and greater apple intake. Plateau pikas show an increase in alpha diversity in winter, influenced by a combination of low winter temperatures, drought, and dietary fiber content. It has also been suggested that the gut microbiota of animals can be influenced by physiological factors, such as reproduction and metabolism, which also show seasonal variation [47]. For example, in the cave rat, the Chao1 index increases between estrus and non-estrus stages in response to changes in gut microbial structure and function [47]. As macaques are characterized by seasonal reproduction [21], their reproductive cycles may affect their gut bacterial diversity.

An LEfSe analysis showed that Firmicutes were significantly enriched in winter, while Bacteroidetes were significantly enriched in summer. In addition, the F/B ratio was markedly higher in winter than in other seasons. A high F/B ratio can reflect weight gain in birds [48] and mammals [49]. Here, we observed that the body weight of rhesus macaques was greater in winter than in the other seasons; however, the difference was not significant. One study showed that the abundance of Bacteroides in rats fed whole apples was lower than that in animals given the control diet [50]. In this study, although the proportion of apples provided in the diet was lower in summer than in winter, the abundance of Bacteroidetes was lower in winter than in summer. This suggested that the significant enrichment of Bacteroidetes observed in summer was related to apple intake. In addition, *S. lutetiensis* was significantly enriched in summer. The isolation of *S. lutetiensis* from the gastrointestinal tract of giant pandas indicates its potential as a probiotic. This species exhibits α-galactosidase and β-glucosidase activities, both of which play crucial roles in the degradation of cellulose and hemicellulose [51,52,53]. In addition, we found that Proteobacteria and Campylobacter were significantly enriched in spring compared to other seasons. The marked enrichment of Campylobacter in the intestinal tract of rhesus macaques was likely related to the increase in the abundance of the genus *Helicobacter*. The phylum Proteobacteria includes various pathogens, such as *Escherichia coli* and *Salmonella*. An increase in the relative abundance of Proteobacteria under conditions of intestinal dysbiosis often leads to diarrhea. Campylobacter spp. within the phylum Campylobacter is closely associated with diarrhea. However, the rhesus monkeys in this study were all healthy, indicating that healthy rhesus monkeys harbor varying levels of Proteobacteria and Campylobacter. The abundance of these bacteria is closely related to the immune status of animals. Diarrhea likely occurs only when the levels of bacteria within these phyla increase abnormally and immunity is compromised. In addition, during winter and spring, temperature changes and other environmental stresses pose thermoregulatory challenges for rhesus macaques. Under such conditions, bacteria that are not predominant in the intestinal tract, such as Proteobacteria and Campylobacter, proliferate, presumably due to trade-offs between adaptation to unfavorable environments and immune regulation. Additionally, beneficial genera, such as *Lactobacillus*, *Limosilactobacillus*, and *Ligilactobacillus*, along with beneficial species, such as *L. johnsonii*, *L. reuteri*, *L. murinus*, and *L. amylovorus*, showed seasonal variations and were significantly enriched in winter. The four beneficial species possess genes encoding mucus-binding proteins, which allow them to adhere to intestinal epithelial cells [22]. This suggests that the effective adhesion of probiotics to the gut is a fundamental mechanism underlying their actions, including the immunomodulation of the host and the inhibition of conditionally pathogenic bacteria. The intestines of macaques harbor beneficial probiotics, which help maintain a stable gut microbiome and alleviate intestinal inflammation. Bacteria of the *Lactobacillus genus*, which are prevalent in rhesus monkeys, are considered potential probiotics that can prevent diarrhea. Moreover, the highest contents of probiotic *Lactobacillus* and related species were observed in the winter, indicating that rhesus monkeys are in a state of constant self-regulating dynamic balance. When conditionally pathogenic bacteria proliferate, beneficial bacteria compete with them to maintain a healthy internal environment.

In this study, the relative abundance of carbohydrate metabolic functions was significantly higher in winter compared to other seasons. This is strongly correlated with the notable enrichment of the core phylum Firmicutes and the core genus *Lactobacillus* during winter, indicating that the seasonal changes in the core bacteria were also closely associated with the seasonal function changes.

The VPA showed that the proportion of fruits and vegetables significantly affected the microbial community, highlighting the critical role that dietary plays in modulating gut microbial communities. For instance, research suggests that approximately 60% of the microbial composition can be rapidly modified through dietary adjustments [54]. In the present study, apples displayed a positive correlation with the relative abundance of the genera *Lactobacillus*, *Limosilactobacillus*, and *Ligilactobacillus* and a negative correlation with the relative abundance of the genus *Prevotella 9*. The abundance of *Prevotella 9* showed a positive correlation with kohlrabi, pumpkin, and white radish with onion. This is indicative of the potential of apples in promoting gut health. Apples are rich in nutrients such as sugars, organic acids, and vitamins, as well as dietary fiber. Studies have shown that the dietary fiber in apples promotes intestinal peristalsis and has laxative effects, while their abundant polyphenols effectively scavenge free radicals [55]. These active ingredients help maintain a dynamic balance in the intestinal flora, thereby exerting health-promoting effects. The authors of [56] demonstrated that consuming two apples a day for 2 weeks increased the abundance of *Bifidobacterium* and *Lactobacillus* spp. in the feces of eight healthy adults. In addition, it has been shown that apple fiber isolate increases both fecal output (weight, frequency) in humans and rats [57] and gut transit time in humans, which may be helpful in constipation. In our study, the relative abundance of *Lactobacillus*, *Limosilactobacillus*, and *Ligilactobacillus* was higher in winter than in the other seasons, likely due to the increased intake of apples during this season. Our findings also suggested that under artificial feeding conditions with fixed compound feeds, different fruit and vegetable compositions and ratios have varying degrees of influence on the composition and function of the intestinal flora of rhesus macaques, which subsequently impacts their metabolism and health. This highlights the importance of a reasonable mix of fruits and vegetables. Their types and proportions can be flexibly adjusted according to seasonal changes to meet the nutritional needs and taste preferences of the macaques. Ensuring the hygiene and quality of these foods is also important for protecting the health of rhesus monkeys.

## 5. Conclusions

Our research indicated that the gut microbiota profiles and essential functionalities of adult rhesus macaques varied significantly across the four seasons. Notably, the alpha diversity, relative abundance of Firmicutes, and Firmicutes-to-Bacteroidetes ratio were markedly elevated during winter compared to the other seasons. Bacteroidetes was significantly enriched in summer, and Proteobacteria and Campylobacter were significantly enriched in spring. The proportions of fruits and vegetables in the diet, but not climatic factors (temperature and humidity), significantly influenced the seasonal changes in the gut microbiota. This indicates that rhesus macaques exhibit some adaptability to changes in temperature and humidity and show that the animal husbandry conditions of the Experimental Animal Centre of the Kunming Institute of Zoology, Chinese Academy of Sciences, can protect the health of rhesus monkeys. This study provides new evidence related to how external environmental factors affect the intestinal environment of rhesus monkeys and offer valuable data for selecting and assessing the quality of fruits and vegetables for macaques.

Nevertheless, this research did not assess the nutritional profiles of fruits and vegetables across various seasons, including components such as dietary fiber and protein content. Consequently, it was unable to furnish direct evidence regarding the influence of these nutrients on the composition of macaque gut microbiota, which still needs further investigation in subsequent studies.

## Figures and Tables

**Figure 1 microorganisms-13-00117-f001:**
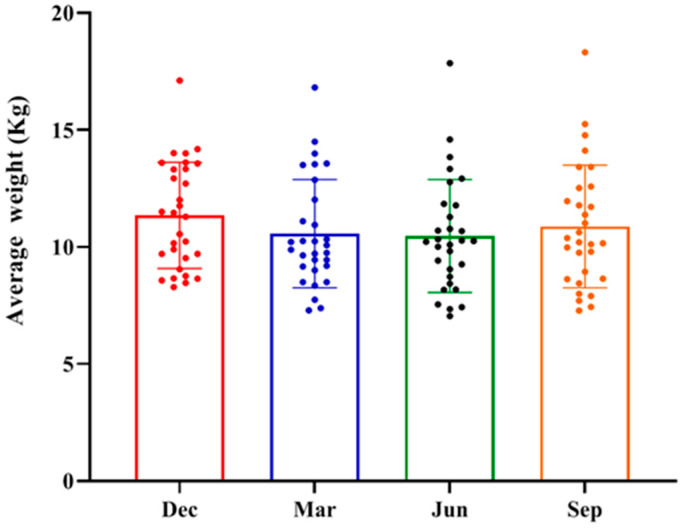
Changes in mean body weight of rhesus monkeys over four seasons.

**Figure 2 microorganisms-13-00117-f002:**
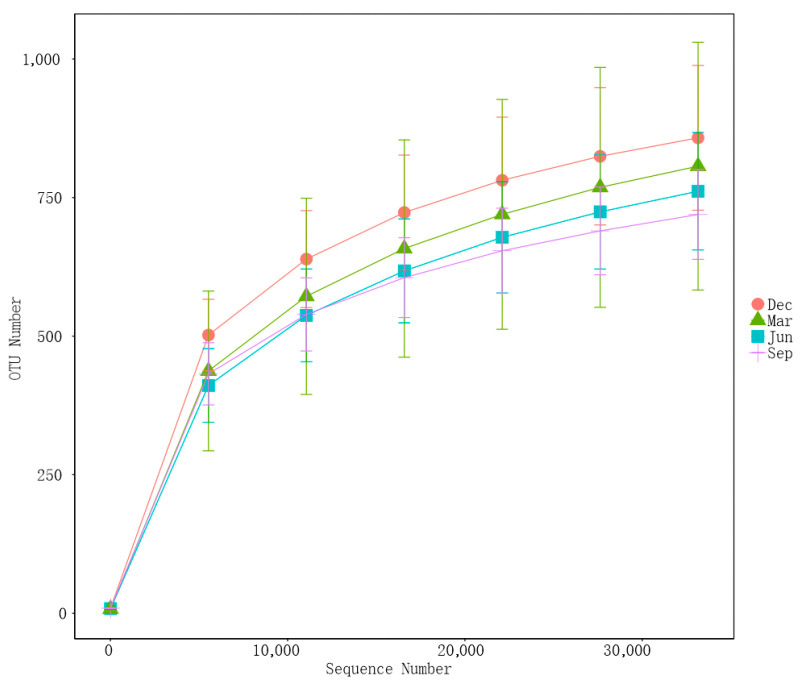
Rarefaction curves of the observed indexes.

**Figure 3 microorganisms-13-00117-f003:**
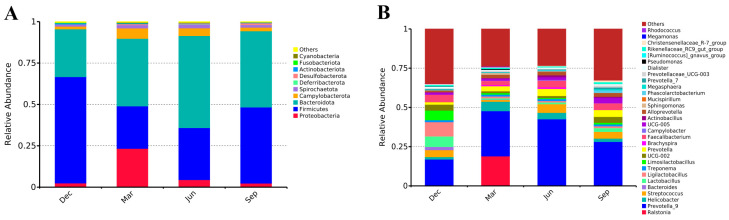
Relative abundance of microbial communities in rhesus monkey fecal samples at the phylum and genus levels. (**A**) Relative abundance of dominant phyla (top 10). (**B**) Relative abundance of dominant genera (top 30).

**Figure 4 microorganisms-13-00117-f004:**
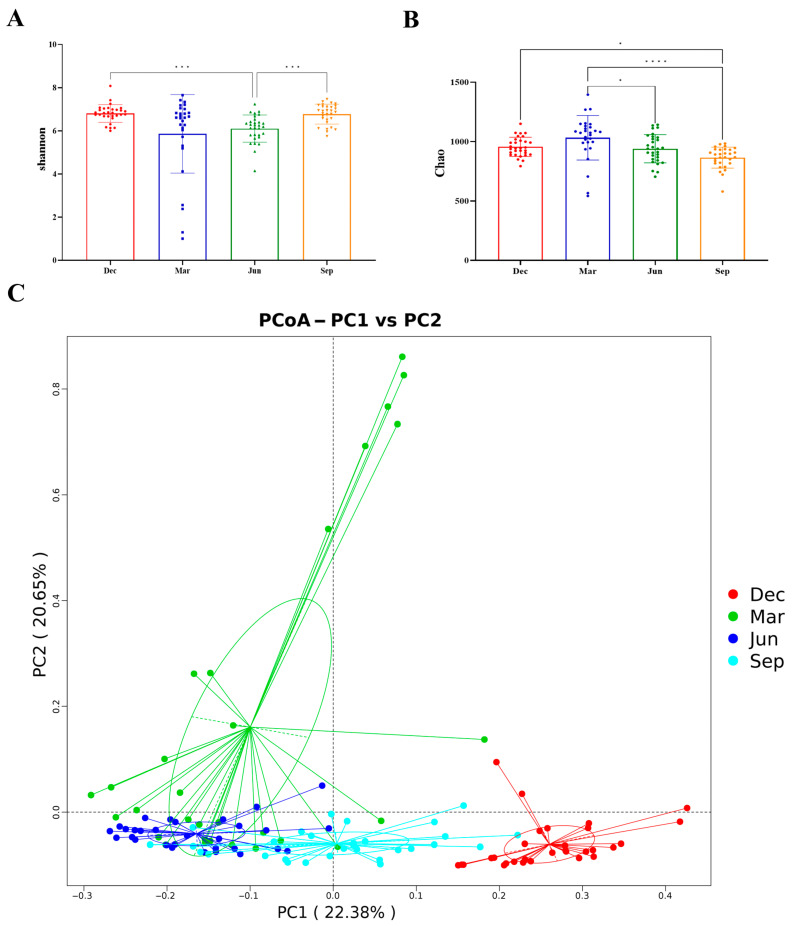
Analysis of the alpha diversity and beta diversity of the gut microbiome of rhesus macaques in different seasons. (**A**) Seasonal differences in the Shannon index. (**B**) Seasonal differences in the Chao1 index. * *p* < 0.05, *** *p* < 0.001, **** *p <* 0.0001 (Wilcoxon rank sum test). (**C**) Principal coordinate analysis (PCoA) of gut microbial composition in rhesus monkeys in different seasons.

**Figure 5 microorganisms-13-00117-f005:**
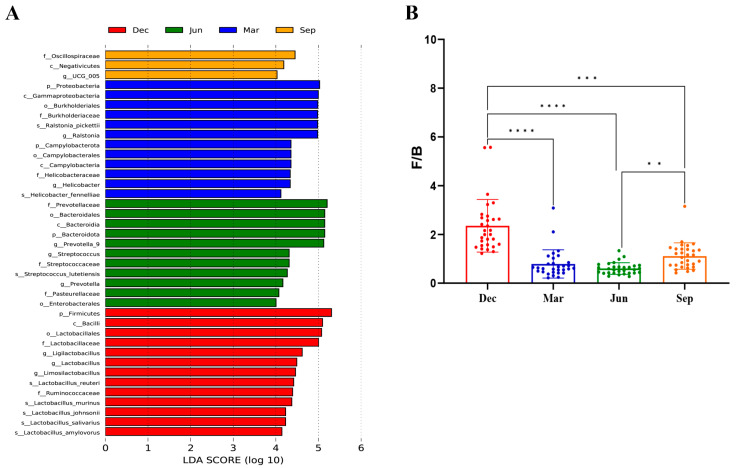
Variations in the bacterial composition of the gut microbiota of rhesus macaques across seasons. (**A**) The phyla, genera, and species present in each season were identified using a linear discriminant analysis (LDA) effect size (LEfSe) analysis (LDA > 4, *p* < 0.05). (**B**) Seasonal differences in the ratio of Firmicutes to Bacteroidetes. ** *p* < 0.01, *** *p* < 0.001, **** *p* < 0.0001 (Wilcoxon rank sum test).

**Figure 6 microorganisms-13-00117-f006:**
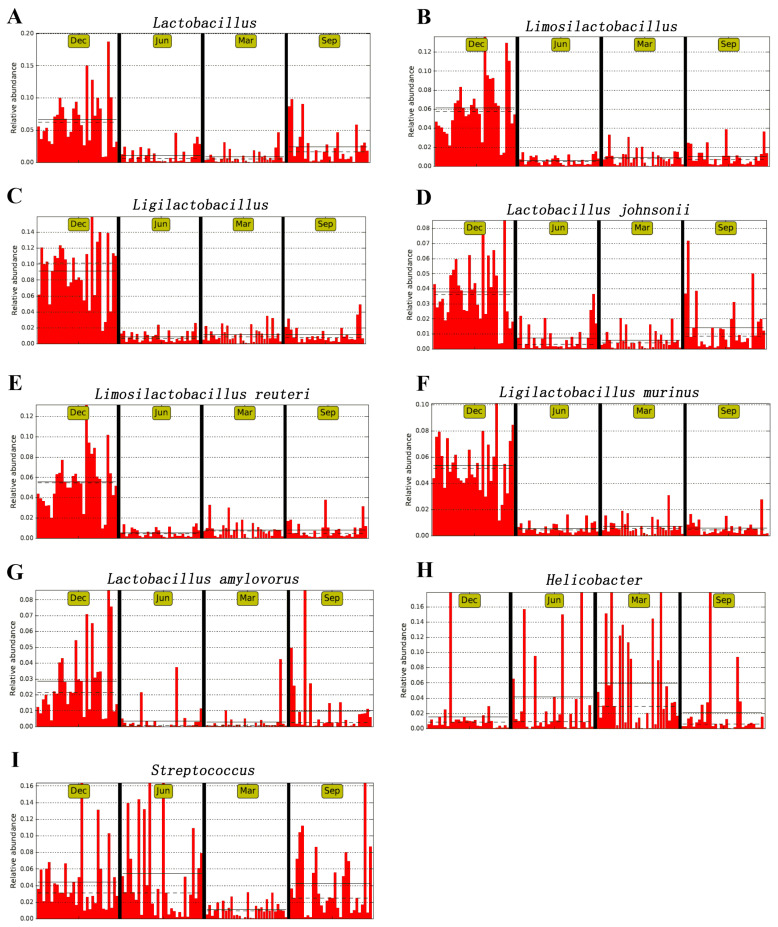
Linear discriminant analysis effect size (LEfSe) analysis of fecal samples in the different seasons. (**A**) *Lactobacillus*, (**B**) *Limosilactobacillus*, (**C**) *Ligilactobacillus*, (**D**) *Lactobacillus johnsonii*, (**E**) *Limosilactobacillus reuteri*, (**F**) *Ligilactobacillus murinus*, (**G**) *Lactobacillus amylovorus*, (**H**) *Helicobacter*, and (**I**) *Streptococcus*.

**Figure 7 microorganisms-13-00117-f007:**
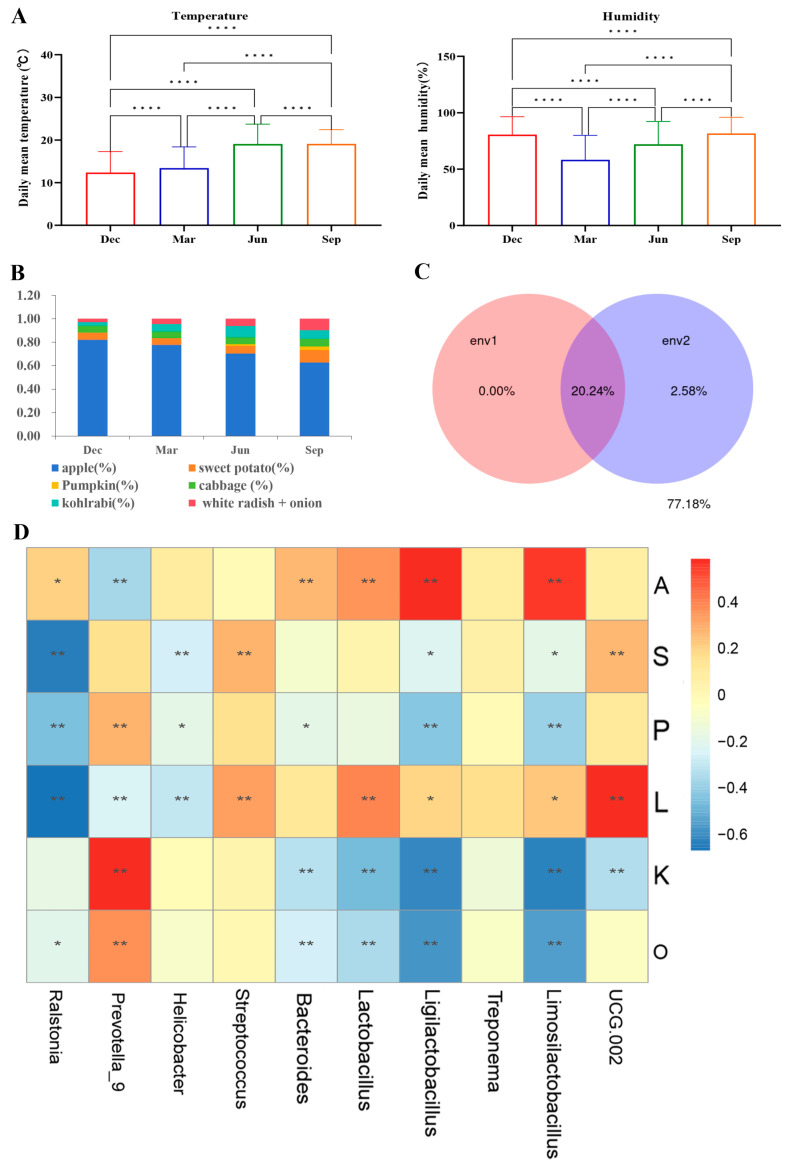
Effects of environmental factors on the seasonal variation in the gut microbiota of rhesus monkeys. (**A**) Environmental factors in the different seasons (mean ± SEM). **** *p* < 0.0001 (Wilcoxon rank sum test). (**B**) The ratio of fruits and vegetables provided in the different seasons. (**C**) Variance partitioning analysis (VPA) of the two types of environmental factors (env2: dietary andenv1: climatic). (**D**) Spearman’s correlations of the types of fruits and vegetables with the dominant genera (* *p* < 0.05, ** *p* < 0.01).

**Figure 8 microorganisms-13-00117-f008:**
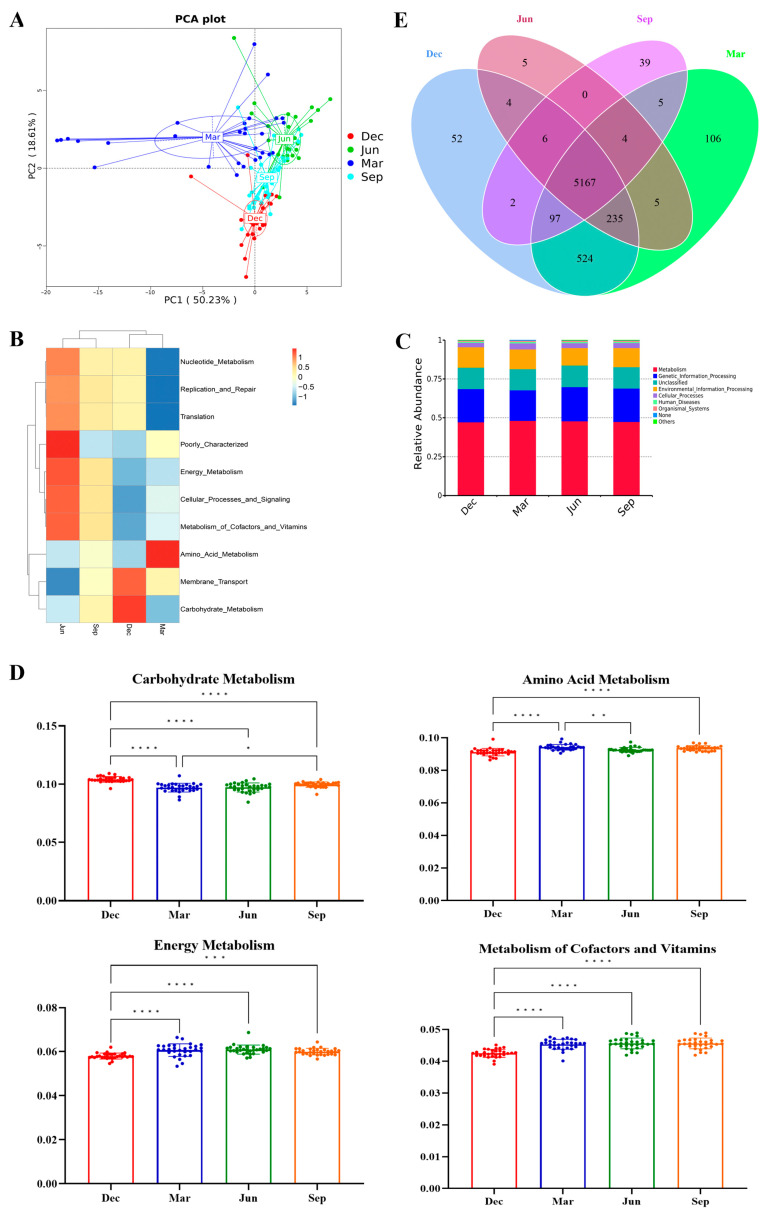
Seasonal differences in gut microbial functions. (**A**) Principal component analysis (PCA) of abundance statistics based on functional annotations against the KEGG database. (**B**) Heatmap of the cluster analysis of functions. (**C**) Relative abundance in level 2 KEGG pathways. (**D**) Seasonal differences in the main functions of the gut microbiota of the rhesus monkeys based on the KEGG database. * *p* < 0.05, ** *p* < 0.01, *** *p* < 0.001, **** *p* < 0.0001 (Wilcoxon rank sum test). (**E**) Venn diagram showing the distribution of the KEGG Orthology (KO) terms among the seasons.

**Table 1 microorganisms-13-00117-t001:** Analysis of similarities (ANOSIM) in the gut microbiota between every two seasons in rhesus monkeys.

	R-Value	*p*-Value
December versus June	0.806	0.001
December versus September	0.5652	0.001
March versus December	0.5899	0.001
March versus September	0.2947	0.001
March versus June	0.2035	0.001
June versus September	0.3278	0.001

**Table 2 microorganisms-13-00117-t002:** Proportions of detected harmful genera.

	December (%)	March (%)	June (%)	September (%)
*Escherichia–Shigella*	17/30 (56.67)	13/30 (43.33)	8/30 (26.67) ^#^	7/30 (23.33) ^#^
*Pseudomonas*	30/30 (100)	24/30 (80) ^‡^	3/30 (10) ^+$^	2/30 (6.67) ^+$^
*Campylobacter*	30/30 (100)	27/30 (90)	28/30 (93.33)	23/30 (76.67) ^&^
*Vibrio*	2/30 (6.67)	9/30 (30)	0/30 (0)	1/30 (3.33)

Escherichia–Shigella: ^#^ represents a significant difference (*p* < 0.05) between December and June, as well as between December and September. Pseudomonas: ^‡^ represents a significant difference (*p* < 0.05) between December and March. ^+^ represents a significant difference (*p* < 0.0001) between December and June, as well as between December and September. ^$^ represents a significant difference (*p* < 0.0001) between March and June, as well as between March and September. Campylobacter: ^&^ represents a significant difference (*p* < 0.05) between December and September.

## Data Availability

The original contributions presented in this study are included in the article. Further inquiries can be directed to the corresponding authors.

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
