# Peer review of "Seasonal Variations in the Structure and Function of the Gut Flora in Adult Male Rhesus Macaques Reared in Outdoor Colonies"

_microorganisms, 2025, doi:10.3390/microorganisms13010117_

Round 1
Reviewer 1 Report
Comments and Suggestions for Authors
In general, the manuscript contains very interesting information and is well analyzed.
The manuscript identifies seasonal changes in the gut microbiota of adult male Rhesus macaques raised in outdoor colonies. However, the authors used stool samples that were representative of the large intestine and rectal microbiota, rather than the small and large intestine. Fecal microbiota is widely used in microbiome research as a proxy for gut health.
December has a higher F/B ratio than other seasons. It refers to higher fermicutes. Fermicutes need and metabolize high carbs and high fat. But there is no differences as shown in Fig 8. What do you explain it?
It is important to draw detailed conclusions on the basis of seasonal feeding. Therefore, you need to write a conclusion and extend it to your next study or apply it in your practice.
Author Response
Point-by-point responses to Reviewer 1’s comments
Reviewer 1:
In general, the manuscript contains very interesting information and is well analyzed.
The manuscript identifies seasonal changes in the gut microbiota of adult male Rhesus macaques raised in outdoor colonies. However, the authors used stool samples that were representative of the large intestine and rectal microbiota, rather than the small and large intestine. Fecal microbiota is widely used in microbiome research as a proxy for gut health.
- December has a higher F/B ratio than other seasons. It refers to higher fermicutes. Fermicutes need and metabolize high carbs and high fat. But there is no differences as shown in Fig 8. What do you explain it?
Response: Thanks a lot for this valuable suggestion. We apologize for any lack of clarity in our previous explanation. Concerning the relationship between Firmicutes and carbohydrate metabolism, we have incorporated a pertinent discussion in the discussion section and emphasized the findings presented in Figure 8D, which indicates that carbohydrate metabolism is significantly elevated in winter compared to other seasons (p < 0.0001). This finding further underscores the critical role of Firmicutes in the degradation of cellulose and hemicellulose, as well as in carbohydrate metabolism and the digestion and absorption of nutrients from dietary sources. In addition, this chapter mainly analyzes the main functions of the top 10 functional relative abundance, and fat metabolism is not among them. However, this does not mean that fat metabolism was not important or not considered in this study. We will further explore and analyze the relationship between fat metabolism and gut microbiome in subsequent studies. In addition, PICRUSt analysis cannot directly reflect the differences between groups from the bar chart.
- It is important to draw detailed conclusions on the basis of seasonal feeding. Therefore, you need to write a conclusion and extend it to your next study or apply it in your practice.
Response: Thanks for the suggestion. In the revised manuscript, we have written a conclusion (lines 575-594).
Lines 575-594: 5. Conclusions
Our research indicated that the gut microbiota profiles and essential functionali-ties of adult rhesus macaques varied significantly across the four seasons. Notably, al-pha diversity, the relative abundance of Firmicutes, and the Firmicutes to Bacteroide-tes ratio were markedly elevated during winter compared to the other seasons. While, Bacteroidetes was significantly enriched in summer, and Proteobacteria and Campyl-obacter were significantly enriched in spring. The proportions of fruits and vegetables in the diet, but not climatic factors (temperature and humidity), significantly influ-enced the seasonal changes in the gut microbiota. This indicates that rhesus macaques exhibit some adaptability to changes in temperature and humidity, and show that the animal husbandry conditions of the Experimental Animal Centre of the Kunming In-stitute of Zoology, Chinese Academy of Sciences can protect the health of rhesus mon-keys. This study provides new evidence relating to how external environmental factors affect the intestinal environment of rhesus monkeys and offer valuable data for se-lecting and assessing the quality of fruits and vegetables for macaques.
Nevertheless, this research did not assess the nutritional profiles of fruits and vegetables across various seasons, including components such as dietary fiber and protein content. Consequently, it was unable to furnish direct evidence regarding the influence of these nutrients on the composition of macaque gut microbiota, which still needs further investigation in subsequent studies.
Reviewer 2 Report
Comments and Suggestions for Authors
microorganisms-3378583-peer-review-v1
This is an interesting research project, providing additional information about the microbiota of Rhesus monkey and how the seasonal conditions can interfere with the specificity of the microbial gut population. The work is interesting; however, the format of the manuscript need additional attention from the authors.
Different parts of the manuscript need attention and need to be more proportional. In addition to the well-presented Results section, discussion is quite basic and deserve more attention from the authors.
Ln34: Do you referring to the Genus Lactobacillus in meaning after changes of taxonomy in 2020? Please, do it is italics. If you are referring in the sense of term before 2020, then use English word "lactobacilli".
Ln35-36: Please, for the mentioned species, use the new names, suggested in 2020 and furter abbreviate them according to the recommendations (https://doi.org/10.1099/ijsem.0.004107.32293557, https://doi.org/10.1163/18762891-20230114).
Ln61: Please, correct to "....[15]. The composition...".
All around the manuscript, please, add intervale before providing the reference. For examples, see Ln 68, 72, 83, etc.
Ln69: human genes or human genome?
Ln135: Please, provide city and country. It is presumed that this is China, but for purpose of the clear text, this information needs to be provided. Please, for all material and equipment, always provide supplier company name, followed by city, state (in case of federal country) in abbreviated way, and name of the country. In following occasions, only name of the company is sufficient. Please, use the headquarters address and not addresses of the local distributions.
Please, in the material and methods avoid providing the results. Body weight of the experimental animals is the results, since you have started with 30 animals, and then you have monitored their body weight. Please, correct this part, and other similar points in the paper.
Ln145: Mycobacterium tuberculosis, Salmonella and Shigella needs to be in italics.
Table 1 can be reduced. Already in the text was stated all this information, except age of the monkeys. Maybe after reduction, table can be move to the Supplementary material.
Figure 1 was already presented clearly in the text regarding fluctuation of the body weight. From this point of view this is repetition of the information.
Paper needs intensive restructuring. Authors extensively provide results in their material and methods sections. Please, correct this. Maybe help from more experience colleagues can be option in reorganization of the manuscript.
Ln262: Streptococcus and helicobacter needs to be in italics. What Streptococcus have you detected? Most Streptococcus in fact are pathogens. If you can provide more information regarding about Streptococcus variety will be good.
Lactobacillus - please, see my previous comments.
Discussion in high extent is repetitive of the previous sections. Please, avoid repeating results in the discussion section. Will be positive if authors can compare observed results with other similar studies and improve discussion section. In current way, discussion is quite basic, compared to the Results, where authors have shown their knowledge and expertise in interpretation of bioinformatics results. Maybe help from more experienced colleagues will be a good option to assist authors in improving discussion section and to be more proportional to rest of the manuscript.
References need to be adjusted to the requirements of the journal.
The manuscript have 34% similarity level, this need additional attention from authors and to be reduced.
Author Response
Dear Reviewer,
Thank you for giving us the opportunity to amend our manuscript and respond to the peer-review. According to your comments, we have modified our manuscript. Below are our detailed and point-by-point responses, which are marked in Blue font in the revised manuscript for easy viewing. We hope that the revisions and our responses will be sufficient to make our manuscript suitable for publication in Microorganisms, and thank you again for your review.
Yours sincerely,
Dongdong Qin
Yunnan University of Chinese Medicine
Point-by-point responses
For Reviewer: 2
Comments and Suggestions for Authors
This is an interesting research project, providing additional information about the microbiota of Rhesus monkey and how the seasonal conditions can interfere with the specificity of the microbial gut population. The work is interesting; however, the format of the manuscript needs additional attention from the authors.
Different parts of the manuscript need attention and need to be more proportional. In addition to the well-presented Results section, discussion is quite basic and deserve more attention from the authors
Response: We are grateful for the suggestion. As you suggested, we have paid more attention to different parts of our manuscript and made more in-depth discussions of the results (lines 430-440; lines 443-455; lines 500-504; lines 509-513; lines 543-551). Also, we have provided the conclusions (lines 575-594).
Lines 430-440: Firmicutes and Bacteroidetes have been identified as the predominant phyla in non-human primates, comprising proportions ranging from 70.50% to 98.30%, respec-tively [4]. Firmicutes facilitate the breakdown of dietary fiber and the conversion of cellulose into volatile fatty acids, thereby improving digestive efficiency and support-ing growth and development [29]. In our investigation, we found that within the top 30 bacterial genera present in the gut microbiota of rhesus macaques, 10 were classi-fied under Firmicutes, which is linked to carbohydrate metabolism as well as cellulose digestion and absorption. Furthermore, 7 genera were categorized within the phylum Bacteroidetes, whose members are primarily involved in the digestion and absorption of proteins and carbohydrates, while also contributing to the maturation of the gas-trointestinal immune system.
Lines 443-455: This dietary pattern resembles the Mediterranean diet (MD) in humans and is also a plant-based diet. The MD is associated with the genus Prevotella and with higher levels of Faecalibacterium prausnitzii, increasing SCFAs (short-chain fatty acids)-producing bacteria [31]. The microbiota plays a crucial role in deriving energy and nutrients from plant-based diets, potentially aiding primates in adapting to new dietary niches as a response to swift environmental changes [32].
At the genus level, the dominant genera in macaques are primarily those that produce SCFAs [33], such as Prevotella 9, Streptococcus, Faecalibacterium, Prevotella, Ligilactobacillus, Lactobacillus, UCG-002, and UCG-005. These SCFAs may play a role in decreasing the incidence of malignancies, such as colorectal cancer, and enhancing cardiometabolic health. Also, these SCFAs can be involved in the regulation of immune function, glucose and lipid metabolism, as well as blood pressure, all of which are linked to the Mediterranean diet (MD) [34, 35].
Lines 500-504: In addition, S. lutetiensis was significantly enriched in summer. The isolation of S. lute-tiensis from the gastrointestinal tract of giant pandas indicates its potential as a probi-otic. This species exhibits α-galactosidase and β-glucosidase activities, both of which play crucial roles in the degradation of cellulose and hemicellulose [51-53].
Lines 509-513: In addition, S. lutetiensis was significantly enriched in summer. The isolation of S. lute-tiensis from the gastrointestinal tract of giant pandas indicates its potential as a probi-otic. This specie exhibits α-galactosidase and β-glucosidase activities, both of which play crucial roles in the degradation of cellulose and hemicellulose [51-53].
Lines 543-551: In this study, the relative abundance of carbohydrate metabolic functions was significantly higher in winter compared to other seasons. This is strongly correlated with the notable enrichment of the core phylum Firmicutes and the core genus Lacto-bacillus during winter, indicating that the seasonal changes in the core bacteria were also closely associated with the seasonal function changes.
The VPA showed that the proportion of fruits and vegetables significantly affect-ed the microbial community, highlighting the critical role that dietary plays in modu-lating gut microbial communities. For instance, research suggests that approximately 60% of the microbial composition can be rapidly modified through dietary adjustments [54].
Lines 575-594: 5. Conclusions
Our research indicated that the gut microbiota profiles and essential functionali-ties of adult rhesus macaques varied significantly across the four seasons. Notably, al-pha diversity, the relative abundance of Firmicutes, and the Firmicutes to Bacteroide-tes ratio were markedly elevated during winter compared to the other seasons. While, Bacteroidetes was significantly enriched in summer, and Proteobacteria and Campyl-obacter were significantly enriched in spring. The proportions of fruits and vegetables in the diet, but not climatic factors (temperature and humidity), significantly influ-enced the seasonal changes in the gut microbiota. This indicates that rhesus macaques exhibit some adaptability to changes in temperature and humidity, and show that the animal husbandry conditions of the Experimental Animal Centre of the Kunming In-stitute of Zoology, Chinese Academy of Sciences can protect the health of rhesus mon-keys. This study provides new evidence relating to how external environmental factors affect the intestinal environment of rhesus monkeys and offer valuable data for se-lecting and assessing the quality of fruits and vegetables for macaques.
Nevertheless, this research did not assess the nutritional profiles of fruits and vegetables across various seasons, including components such as dietary fiber and protein content. Consequently, it was unable to furnish direct evidence regarding the influence of these nutrients on the composition of macaque gut microbiota, which still needs further investigation in subsequent studies.
Ln34: Do you referring to the Genus Lactobacillus in meaning after changes of taxonomy in 2020? Please, do it is italics. If you are referring in the sense of term before 2020, then use English word "lactobacilli"
Response: Thanks for the suggestion. As you suggested, we referred to the Genus Lactobacillus in meaning after changes of taxonomy in 2020, and have changed the Lactobacillus into italics. We have also checked the entire text, and changed the Lactobacillus into italics wherever it appeared.
Ln35-36: Please, for the mentioned species, use the new names, suggested in 2020 and further abbreviate them according to the recommendations (https://doi.org/10.1099/ijsem.0.004107.32293557, https://doi.org/10.1163/18762891-20230114)
Response: Thanks for the suggestion. As you suggested, we have checked the entire text, and confirmed that the mentioned species have used the new names, which were suggested in 2020. In the abstract, we have used the full names of the species for the first mention. Subsequently, we have abbreviated them according to your recommendations.. Below is the updated information.
Lactobacillus johnsonii, abbreviated L. johnsonii, and
Lactobacillus reuteri, abbreviated L. reuteri,
Lactobacillus murinus, abbreviated L. murinus
Lactobacillus amylovorus, abbreviated L. amylovorus.
Ln61: Please, correct to "....[15]. The composition...".
All around the manuscript, please, add intervale before providing the reference. For examples, see Ln 68, 72, 83, etc.
Response: Thanks for the suggestion. We apologize for our carelessness. As you suggested, we have added the intervale before providing the reference and corrected any errors that occurred.
Ln69: human genes or human genome?
Response: Thanks for the suggestion. This refers to the human genome.
Ln135: Please, provide city and country. It is presumed that this is China, but for purpose of the clear text, this information needs to be provided. Please, for all material and equipment, always provide supplier company name, followed by city, state (in case of federal country) in abbreviated way, and name of the country. In following occasions, only name of the company is sufficient. Please, use the headquarters address and not addresses of the local distributions.
Response: Thanks for the suggestion. As you suggested, we have made corresponding corrections and provided the city and country. For all first-appearance materials and equipment, we also provided supplier company name (headquarters address), followed by city, state, and name of the country. In following occasions, we used only name of the company.
Please, in the material and methods avoid providing the results. Body weight of the experimental animals is the results, since you have started with 30 animals, and then you have monitored their body weight. Please, correct this part, and other similar points in the paper.
Response: Thanks for the suggestion. As you suggested, in the revised manuscript, we have deleted all the information about the results in the material and methods. The results of body weights have been moved to the result section “3.1. Animal Demographics” (lines 243-249).
Lines 243-249: 3.1. Animal demographics
Thirty healthy adult male rhesus macaques are negative for seven pathogens (TB, SRV, STLV, SIV, BV, Salmonella and Shigella). These animals have never had diarrhea, and have not used any medication or probiotics. Their ages range from 10 to 15 years (11.50 ± 1.78). The mean body weight was 11.35 ± 2.27 kg in winter, 10.57 ± 2.31 kg in spring, 10.47 ± 2.42 kg in summer, and 10.87 ± 2.62 kg in autumn. No significant dif-ferences were found in body weights among the four seasons (p > 0.05) (Figure 1).
Ln145: Mycobacterium tuberculosis, Salmonella and Shigella needs to be in italics.
Response: Thanks for the suggestion. As you referred in the above comment, we have changed Mycobacterium tuberculosis, Salmonella, and Shigella into italics.
Table 1 can be reduced. Already in the text was stated all this information, except age of the monkeys. Maybe after reduction, table can be move to the Supplementary material.
Response: Thanks for the suggestion. We have carefully considered your suggestion and decided to delete this table. All information that previously appeared in this table has been included in the result section “3.1. Animal Demographics” (lines 243-249).
Lines 243-249: 3.1. Animal demographics
Thirty healthy adult male rhesus macaques are negative for seven pathogens (TB, SRV, STLV, SIV, BV, Salmonella and Shigella). These animals have never had diarrhea, and have not used any medication or probiotics. Their ages range from 10 to 15 years (11.50 ± 1.78). The mean body weight was 11.35 ± 2.27 kg in winter, 10.57 ± 2.31 kg in spring, 10.47 ± 2.42 kg in summer, and 10.87 ± 2.62 kg in autumn. No significant differences were found in body weights among the four seasons (p > 0.05) (Figure 1).
Figure 1 was already presented clearly in the text regarding fluctuation of the body weight. From this point of view this is repetition of the information.
Response: Thanks for the suggestion. As you suggested, we have excluded the repetitive data of the body weight in the section of 2. Materials and Methods, which has been presented clearly in the section of Results and Figure 1.
Paper needs intensive restructuring. Authors extensively provide results in their material and methods section, correct this. Maybe help from more experience colleagues can be option in reorganization of tons. Pleasehe manuscript.
Response: Thanks for the constructive suggestion. Following your suggestion, we have reconducted our manuscript, deleted the information about results in the section of material and methods. Specifically, we have divided the original section “2.1. Experimental Animals” into two distinct parts: “2.1. Animal Selection” and “2.2. Animal Housing”, in order to more clearly explain the screening criteria and housing conditions of the experimental animals. Additionally, we have removed the original Table 1 and have appropriately integrated these data into the "3.1. Animal Demographics" (lines 243-249).
Ln262: Streptococcus and helicobacter need to be in italics. What Streptococcus have you detected? Most Streptococcus in fact are pathogens. If you can provide more information regarding about Streptococcus variety will be good.
Response: Thanks for the suggestion. We have changed Streptococcus and Helicobacter into italics, and we have ensured that the format of strains and genera in the full text conforms to the specifications. Regarding the issue of the Streptococcus species you mentioned, we found Streptococcus lutetiensis was significantly enriched in summer. Although most Streptococcus species are pathogenic, research has reported the isolation of S. lutetiensis from the gastrointestinal tract of giant pandas indicates its potential as a probiotic. This specie exhibits α-galactosidase and β-glucosidase activities, both of which play crucial roles in the degradation of cellulose and hemicellulose [51-53]. Also, we have added this part to the revised manuscript (lines 509-513).
Lines 509-513: In addition, S. lutetiensis was significantly enriched in summer. The isolation of S. lutetiensis from the gastrointestinal tract of giant pandas indicates its potential as a probiotic. This specie exhibits α-galactosidase and β-glucosidase activities, both of which play crucial roles in the degradation of cellulose and hemicellulose [51-53].
Lactobacillus - please, see my previous comments.
Response: Thanks for the suggestion. We have checked the entire text and changed the Lactobacillus into italics wherever it appeared.
Discussion in high extent is repetitive of the previous sections. Please, avoid repeating results in the discussion section. Will be positive if authors can compare observed results with other similar studies and improve discussion section. In current way, discussion is quite basic, compared to the Results, where authors have shown their knowledge and expertise in interpretation of bioinformatics results. Maybe help from more experienced colleagues will be a good option to assist authors in improving discussion section and to be more proportional to rest of the manuscript.
Response: Thanks for the suggestion. As you comment above, our previous discussion was quite basic, compared to the Results. In the revised manuscript, we have consulted experienced colleagues and tried our best to improve the discussion section, making it more proportional to rest of the manuscript (lines 430-440; lines 443-455; lines 500-504; lines 509-513; lines 543-551). Also, we have provided the conclusions (lines 575-594).
References need to be adjusted to the requirements of the journal.
Response: Thanks for the suggestion. We have adjusted the references to comply with the requirements of the journal.
The manuscript have 34% similarity level, this need additional attention from authors and to be reduced.
Response: Thanks for your concern. As you suggested, we have paid more attention to this and tried our best to reduce the similarity level as much as possible. The current similarity index is 27% through iThenticate detection, mainly some repetition in the section of Materials and Methods.
Round 2
Reviewer 2 Report
Comments and Suggestions for Authors
Authors have improved the manuscript; however, several parts still need attentions form the authors.
Ln35: Limosilactobacillus reuteri, Ligilactobacillus murinus
Ln323, 338, 530 etc.: for the abbreviations, please, follow the recommendations from https://doi.org/10.1163/18762891-20230114
In figure 6, correct the name of former Lactobacillus to the new names. Genera and Species names need to be in italics.
Table 2: Genus names needs to be in italics.
References are still not into the style recommended by the MDPI/Microorganisms.
Author Response
Point-by-point responses
Authors have improved the manuscript; however, several parts still need attentions from the authors.
Ln35: Limosilactobacillus reuteri, Ligilactobacillus murinus
Ln323, 338, 530 etc.: for the abbreviations, please, follow the recommendations from https://doi.org/10.1163/18762891-20230114IF: 5.4 Q1
Response: Thanks for the suggestion. As you suggested, we have made the corresponding revisions to the abstract and the figure legend in Figure 6. Specifically, “Lactobacillus reuteri” has been revised to “Limosilactobacillus reuteri” and “Lactobacillus murinus” has been revised to “Ligilactobacillus murinus”, which are marked in green font in the revised manuscript. For “Lactobacillus johnsonii” and “Lactobacillus amylovorus”, we have maintained the original terminology since there have been no alterations in their classification within the most recent taxonomy. Furthermore, we have confirmed that the nomenclature of these species in the manuscript complies with the appropriate abbreviations as outlined in your suggested publication.
In figure 6, correct the name of former Lactobacillus to the new names. Genera and Species names need to be in italics.
Response: Thanks for the suggestion. As you suggested, we have corrected the name of former Lactobacillus to the new names. Also, Genera and Species names have been italicized.
Table 2: Genus names need to be in italics.
Response: Thanks for the suggestion. As you suggested, Genus names have been italicized in Table 2.
References are still not into the style recommended by the MDPI/Microorganisms.
Response: Thanks for the suggestion. In accordance with the Full Reference Formatting Guidelines provided by MDPI, we utilized an EndNote template specifically designed for the MDPI Chicago Style to modify the references accordingly.
